# Association between Liver Stiffness and Liver-Related Events in HCV-Infected Patients after Successful Treatment with Direct-Acting Antivirals

**DOI:** 10.3390/medicina59030602

**Published:** 2023-03-18

**Authors:** Napas Rodprasert, Tinn Hongboontry, Chitchanok Cherdchoochart, Roongruedee Chaiteerakij

**Affiliations:** 1Division of Gastroenterology, Department of Medicine, King Chulalongkorn Memorial Hospital, Bangkok 10330, Thailand; 2Department of Medicine, Faculty of Medicine, Chulalongkorn University, Bangkok 10330, Thailand; 3Center of Excellence for Innovation and Endoscopy in Gastrointestinal Oncology, Division of Gastroenterology, Faculty of Medicine, Chulalongkorn University, Bangkok 10330, Thailand

**Keywords:** chronic hepatitis C infection, transient elastography, liver fibrosis, fibrosis score, liver steatosis, liver-related complications, liver cancer

## Abstract

*Background and Objectives*: Direct-acting antivirals (DAAs) are highly effective for the treatment of chronic hepatitis C virus (HCV) infection, but the risk of liver-related events and hepatocellular carcinoma (HCC) remains after successful therapy. We aimed to evaluate post-treatment changes in liver stiffness (LS) and identify a cut-off LS value for predicting such events in chronic HCV-infected patients receiving DAA. *Materials and Methods*: A total of 185 patients who had achieved sustained virologic response (SVR) after DAA therapy were included. Baseline characteristics and laboratory results were retrospectively abstracted. LS was measured by transient elastography at baseline, 12, 24, 48, and 96 weeks after SVR. FIB-4 index was assessed at baseline and 48 weeks after SVR. Development of liver-related events (hepatocellular carcinoma (HCC), portal-hypertension-related decompensation, listing for transplantation, and mortality) after SVR were identified. The association between liver fibrosis and the occurrence of liver-related events was analyzed using Cox regression analysis. *Results*: Significant differences in LS values were observed between baseline and 24, 48, 72, and 96 weeks after SVR. FIB-4 index at 48 weeks after SVR was significantly lower than the FIB-4 index at baseline. During the 41.6-month follow-up time, the incidence rates of all liver-related events and HCC were 2.36 and 1.17 per 100 person-years, respectively. Age, LS ≥8 kPa, and FIB-4 ≥1.35 at 48 weeks post-SVR were significantly associated with the occurrence of any liver-related events. By multivariate analysis, LS ≥8 kPa at 48 weeks post-SVR remained significantly associated with any liver-related events, with an adjusted hazard ratio (95%CI) of 5.04 (1.01–25.26), *p* = 0.049. *Conclusions*: Despite a significant reduction in LS after SVR, patients with LS ≥8 kPa at 48 weeks after SVR should be regularly monitored for liver-related complications, particularly HCC development.

## 1. Introduction

Chronic hepatitis C virus (HCV) infection is one of the common causes of chronic liver disease [1]. In 2019, an estimated 58 million people were chronically infected with HCV, resulting in 290,000 deaths worldwide. If left untreated, chronic HCV infection causes persistent hepatic inflammation, leading to progressive hepatic fibrosis, cirrhosis, and hepatocellular carcinoma (HCC) [2,3].

Direct-acting antivirals (DAAs) are currently a standard treatment for chronic HCV infection. DAAs specifically inhibit the synthesis of HCV-encoded proteins necessary for viral replication. DAAs have higher rates of sustained virologic response (SVR) and more favorable safety profiles than traditional interferon-based regimens [4,5]. In a study of 102 HCV-infected patients treated with DAA, DAA therapy was significantly associated with liver fibrosis regression [6]. Another study consistently reported that patients who achieved SVR with DAA treatment had significantly reduced liver fibrosis measured by transient elastography (TE), i.e., from a median liver stiffness (LS) of 8.3 kilopascals (kPa) at baseline to 5.4 kPa at 48 weeks after the end of treatment [7]. Achieving SVR was found to be significantly associated with lower risks of all-cause mortality over an 8.4-year follow-up period [8]. Patients with SVR had a markedly lower 10-year cumulative HCC incidence rate than those without SVR, 5.1% vs. 21.8%, respectively [8].

Despite the high efficacy of DAAs, liver fibrosis regression after SVR is not always guaranteed, and the risk of liver-related events, particularly hepatocellular carcinoma (HCC), is not completely eliminated, with an increasing number of reports of HCC development following successful therapy. At 96 weeks after successful DAA treatment, 17% of patients had progressive hepatic fibrosis [9]. In another observational study, the progression of liver fibrosis occurred in 12.5% of patients achieving SVR post-DAA therapy [10]. Notably, the incidence rate of HCC was the highest among patients with fibrosis progression (6.17/100 patient-years) compared to patients with stable fibrosis (1.09/100 patient-years) and those with fibrosis regression (0.75/100 patient-years) [10]. These findings supported an observation in a previous cohort study showing that patients with progressive liver fibrosis after SVR had a significantly higher rate of HCC development than those with fibrosis regression or stability (33% vs. 4% at 5 years, *p* < 0.001) [11]. After achieving SVR, the annual incidence of HCC was reported to be 1.5 per 100 patient-years in patients with HCV-associated compensated advanced chronic liver disease, and the incidence increased to 1.97 per 100 patient-years in patients with HCV-related cirrhosis [12,13]. These findings highlight the importance of long-term evaluation and monitoring of liver fibrosis in determining the risk of HCC after SVR.

A liver biopsy is the gold standard for diagnosing and staging liver fibrosis. However, it is invasive, with potentially fatal complications and sampling error [14]. As a result, several non-invasive tests, such as the TE and Fibrosis-4 (FIB-4) index, which indirectly reflects the degree of liver fibrosis, have been used to assess liver fibrosis [15,16]. According to international guidelines, following SVR, cirrhotic patients should have an ultrasound examination every 6 months for HCC surveillance, but monitoring for liver-related events or HCC occurrence is currently not recommended for non-cirrhotic patients [17,18]. Although TE and FIB-4 index is currently used to evaluate liver fibrosis, it has not yet been recommended for the post-treatment monitoring of chronic HCV patients due to the lack of consensus on a cut-off value for liver fibrosis that predicts liver-related events in patients who have achieved SVR [19]. This study aimed to evaluate the post-treatment changes in liver stiffness (LS) up to 96 weeks after SVR and to determine the cut-off of LS associated with liver-related events, particularly HCC development, for chronic HCV-infected patients after SVR with DAA therapy.

## 2. Materials and Methods

### 2.1. Patients

All patients with chronic HCV infection who received DAA therapy at the King Chulalongkorn Memorial Hospital, Bangkok, Thailand, between January 2016 and December 2020 were identified (n = 864). The inclusion criteria were as follows: (i) aged 18-70 years, (ii) having completed the DAA regimen and achieved SVR, and (iii) having LS measurement by TE prior to starting DAA and at least one of the following time points: 24, 48, 72, and 96 weeks after achieving SVR. Patients were excluded if they had concomitant etiologies of chronic liver disease, e.g., chronic hepatitis B or human immunodeficiency virus infection, or if they had been diagnosed with HCC or terminal stage of any disease such as end-stage renal disease.

### 2.2. HCV Treatment

The regimen and duration of DAA therapy for each patient were determined by the American Association for the Study of Liver Diseases (AASLD) guidelines, along with the expert opinion of hepatologists at our institution. During the study period, several sofosbuvir-based DAA regimens were available and prescribed to patients, including sofosbuvir/daclatasvir, sofosbuvir/ledipasvir, sofosbuvir/velpatasvir, and sofosbuvir/interferon. Ribavirin was given to patients with cirrhosis and those who had previously failed therapy.

### 2.3. Data Collection

We retrospectively abstracted clinical and laboratory data prior to starting DAA treatment from electronic medical records, including age, gender, body mass index (BMI), comorbidities, presence of cirrhosis, history of HCV treatment (naïve vs. experienced treatment), liver function test, complete blood count, HCV RNA viral load, and genotype. HCV RNA viral load was measured at baseline and 12 weeks after the end of treatment by real-time PCR. The HCV genotype was determined using the linear array method. SVR was defined as undetectable HCV RNA in the blood at 12 weeks or more after completing the DAA regimen. The diagnosis of cirrhosis was made by liver histology and/or radiologic evidence of small nodular surface liver with portal hypertension (e.g., collateral vessels and splenomegaly).

LS was measured before starting DAA and after completing the regimen by TE with Fibrocan (FibroScan^®^ 502 Touch, Echosens, Paris, France). Liver steatosis was measured by controlled attenuation parameter (CAP) with Fibroscan. The median LS and CAP values were expressed in kPa and decibels per meter (dB/m), respectively. Details of the technical background and examination procedure have been previously described [20]. The FIB-4 index was calculated using Sterling’s formula [16].

The follow-up period began at the date of SVR and ended when any liver-related events occurred. The occurrence of liver-related events was defined as death from any causes (liver and non-liver-related mortality), being listed for liver transplantation, being diagnosed with HCC, and having portal hypertension-related liver decompensation. If no event occurred, the follow-up ended in December 2020. The diagnosis of HCC was made using dynamic contrast radiologic criteria or histology when available [21]. Portal hypertension-related liver decompensation was defined as an episode of either ascites development, variceal bleeding, or overt hepatic encephalopathy. Death caused by portal hypertension-related liver decompensation or HCC was considered liver-related mortality, while death from other causes was classified as non-liver-related mortality.

### 2.4. Statistical Analysis

Quantitative variables were expressed as mean with standard deviation (SD) or median with range or interquartile range (IQR) as appropriate. Qualitative variables were expressed as absolute frequencies and percentages. The difference between LS and CAP at baseline and LS and CAP at 24, 48, 72, and 96 weeks after the end of DAA treatment was calculated to determine the magnitude of changes in LS and CAP after successful DDA therapy. The difference in the FIB-4 index between baseline and 48 weeks after SVR was also assessed. The incidence of liver-related events was counted and displayed as a rate per 100 person-years. In the case that a patient experienced more than one liver-related event, the first event was used for statistical analysis. A Cox regression analysis was applied to determine the association of the LS value and the FIB-4 index at various time points and the occurrence of liver-related events. Other factors associated with liver-related events were also identified. Factors significantly associated with an incident liver-related event identified in univariate analysis were included in multivariate analysis adjusted for age and gender. All statistical analyses were performed by IBM SPSS Statistics version 28.0.0 (IBM Corp, Armonk, NY, USA). A *p*-value of 0.05 was considered significant.

## 3. Results

### 3.1. Baseline Characteristics of the Study Cohort

A total of 185 chronic HCV-infected patients met the selection criteria and were included in the analysis. Table 1 summarizes the baseline characteristics of the patient cohort. There were 115 (62.2%) males, with a mean age (± SD) of 56.1 ± 10.9 years. Cirrhosis was present in 80 (43.2%) patients prior to DAA treatment. Genotype 1 was the most common genotype (n = 100, 54%), followed by genotypes 3 and 6. Most patients (n = 130, 70.3%) were naïve to HCV treatments. Sofosbuvir/daclatasvir/ribavirin was the most commonly prescribed DDA regimen in this cohort (n = 53, 28.7%). The percentages of patients receiving each DAA regimen are shown in Table 1. Nearly all patients (n = 175, 94.6%) were prescribed a 12-week DAA regimen.

### 3.2. Liver Stiffness at Baseline and at 24, 48, 72, and 96 Weeks after SVR

Prior to treatment, the median LS value of the entire cohort was 11.8 (range: 7.4–21.2) kPa. During follow-up, LS measurements were taken at 24, 48, 72, and 96 weeks after SVR in 142, 34, 41, and 48 patients, respectively. Table 2 displays the LS values at each follow-up time point. LS value significantly decreased from 12 kPa at baseline to 8.5 kPa at 24 weeks after SVR, *p* < 0.001. The LS value progressively declined to 5.8 kPa at 48 weeks post-SVR, *p* < 0.001. The LS decreased to a lesser extent at 72 weeks and 96 weeks post-SVR, but the values remained statistically significant difference from the baseline value (Table 2).

### 3.3. Liver Steatosis at Baseline and at 24, 48, 72, and 96 Weeks after SVR

Of the 169 patients who had CAP value available before DAA treatment, the median value was 230 (IQR: 204-259) dB/m. There were 136, 32, 36, and 36 patients who had CAP values at baseline and 24, 48, 72, and 96 weeks after SVR, respectively. No significant change was observed between CAP at baseline and 24, 48, 72, and 96 weeks after SVR (Table 2).

### 3.4. FIB-4 Index at Baseline and 48 Weeks after SVR

Similar to the changes in LS value after SVR, a significant decrease in the FIB-4 index was observed. The median FIB-4 index was 2.08 (IQR: 1.21–4.00) at baseline and 0.55 (IQR: 0.05–2.03) at 48 weeks after SVR, *p* < 0.001.

### 3.5. Liver-Related Events after SVR with DAA Therapy

During the mean follow-up time of 41.6 months after SVR, the incidence rate of any liver-related events was 2.36 per 100 person-years (Table 3). Six patients developed HCC, accounting for the 1.17 per 100 person-years HCC incidence rate (Table 4). All six patients had radiologic examinations to confirm the absence of HCC prior to HCV treatment. Two patients experienced liver decompensation as a result of portal hypertension (0.39 per 100 person-years). One had variceal bleeding and ascites on the same visit, and the other had variceal bleeding only. One patient had been listed for liver transplantation (0.19 per 100 person-years). None of the patients had died from liver-related causes, while five died from non-liver-related causes (0.95 per 100 person-years), i.e., two from COVID-19 infection, two from sepsis, and one from lung cancer.

### 3.6. Factors Associated with Liver-Related Events

We found that the LS values at 24 and 72 weeks after SVR were not significantly different from the value at 48 weeks after SVR (*p* = 0.18 and 0.86). To maximize the power of this analysis, we used their LS values at 24 or 72 weeks post-SVR for those who did not have an LS value at 48 weeks post-SVR available, i.e., LS value at 48 ± 24 weeks, yielding a total of 169 patients for this analysis.

In univariate analysis, age, LS value ≥ 8 kPa and FIB-4 index ≥ 1.35 at 48 weeks after SVR were significantly associated with any liver-related events in univariate analysis, with HRs (95% CI) of 1.08 (1.02–1.15), 5.68 (1.23–26.31), and 5.18 (1.10–24.32), *p* = 0.008, 0.026, and 0.037, respectively (Table 5).

Because the LS value and the FIB-4 index reflect the degree of liver fibrosis, we performed multivariate analyses of 3 models to identify independent factors associated with liver-related outcomes. When age and sex were controlled for, the LS value of ≥8 kPa but not the FIB-4 index at 48 weeks after SVR were significantly associated with liver-related events (Table 6, Model 1 and 2). In the third model in which LS value and FIB-4 index were covariates, only the LS value remained independently associated with the occurrence of any liver-related events, with an adjusted HR (95% CI) of 5.04 (1.01–25.26), *p* = 0.049 (Table 6). Figure 1 depicts the cumulative incidence of any liver-related events based on the LS value at 48 weeks after SVR.

## 4. Discussion

This study tracked the progression of liver fibrosis and identified the LS value as a predictor of liver-related events in HCV-infected patients who had received successful DAA therapy. Patients who achieved SVR after DAA therapy had a significant decrease in LS. Despite the regression of fibrosis, the incidence of unfavorable clinical outcomes, including HCC, after SVR remained. Accordingly, our findings emphasized the importance of monitoring LS after completion of DAA therapy, particularly at 48 ± 24 weeks, to determine the need for long-term follow-up for HCC surveillance and monitoring of liver decompensation.

The degree of liver fibrosis is known to be significantly reduced after achieving SVR [7,8,22]. In our study, patients who achieved SVR had a significant decrease in LS, with the greatest decrease of 3.5 pKa at 24 weeks, and the improvement of LS was modest at 72-96 weeks post-SVR. This finding was consistent with a previous study, which reported that patients who achieved SVR had significantly reduced LS values, with the greatest difference between baseline and 24 weeks, but no significant difference between 24 and 48 weeks [7]. A systematic review and meta-analysis of 24 studies which estimated the weighted mean difference in LS reported a significant decrease in LS of approximately 3.1 kPa from baseline to 6–48 weeks after achieving SVR, in contrast to an unchanged LS in those who did not achieve SVR [23].

Non-invasive tests have been used to longitudinally follow patients with HCV infection and to assess the effectiveness of antiviral treatment. A previous retrospective study found that the FIB-4 index improved significantly at 12 weeks after SVR in patients treated with a sofosbuvir-based regimen [24], with the mean ± SD of the FIB-4 index decreasing from 2.7 ± 2.2 to 2.0 ± 1.6 (*p* < 0.01). Another retrospective study of patients treated with paritaprevir/ritonavir/ombitasvir plus dasabuvir found that the median (IQR) of the FIB-4 index decreased from 3.6 (2.3–5.4) to 2.7 (2.0–3.8) at 12 weeks after completion of treatment (22). Similar to these studies, our cohort had a significantly lower FIB-4 index 48 weeks after successful DAA treatment.

Despite a significant reduction in the LS value, our study found that the incidence rates of all liver-related events and HCC were 2.36 per 100 person-years, and 1.17 per 100 patient-years, respectively, during the mean follow-up duration of 41 months, or 2.7 years, after SVR. All patients who developed HCC after SVR had a decrease in LS between baseline and 48 weeks after SVR, and 5 of 6 patients had cirrhosis prior to treatment. Previous studies reported similar HCC incidence rates ranging from 0.90 to 1.5 per 100 patient-years, with a higher HCC incidence rate in patients with cirrhosis who achieved SVR versus those without cirrhosis [12,13,25]. Another study reported the incidence of HCC, all-cause mortality, and liver-related mortality of 5.6%, 2.4%, and 10.4% in 125 patients who had SVR after HCV treatment [8]. No liver-related mortality was detected in our cohort, possibly due to the relatively short follow-up period. As reported in a long-term follow-up study of patients with HCV-related cirrhosis, the risk of HCC, liver decompensation, and liver-related mortality persisted for up to 8 years after SVR [26].

Most importantly, we found that an LS value of ≥8 kPa at 48 weeks post-SVR was significantly associated with poor liver-related outcomes. LS <8 kPa is currently recommended as the cut-off for ruling out advanced fibrosis in patients with alcoholic liver disease and non-alcoholic fatty liver disease [19]. However, due to inconsistencies in previous studies’ results, using LS cut-off values by TE for determining fibrosis regression after SVR in HCV-infected patients has not been recommended [19]. In a study which compared pre- and post-treatment LS measured by TE and liver biopsy, the LS cut-off at 12 kPa was found to be suboptimal with 95% specificity but 61% sensitivity for diagnosing cirrhosis after SVR [27]. By contrast, another study in patients with recurrent HCV infection after liver transplantation demonstrated that LS by TE one year post-SVR accurately predicted the presence of advanced fibrosis, with the best cut-off values at 10.6 kPa and 14 kPa, respectively, to rule out and rule in advanced fibrosis [28]. Moreover, many studies attempted to identify the relationship between LS (before, after and changing after DAA treatment) and the development of HCC. Table 7 summarizes observational studies that assessed changes in LS after SVR and their association with the incidence of HCC in chronic HCV-infected patients. Most comparable to our study, a study in a population of compensated advanced chronic liver disease patients who achieved SVR after DAA reported that LS < 10 kPa at follow-up was a predictor of a lower HCC incidence rate at <1 per 100 patient-years [12]. As suggested by the guideline, the significant LS decrease observed after SVR highlights the need for lower cut-off values to be defined and validated [19]. Thus, our study provides an LS value that could be further investigated as a cut-off for predicting poor long-term liver-related outcomes in chronic HCV-infected patients following SVR.

It remains controversial whether the baseline LS value prior to DAA therapy and the degree of LS improvement is related to the progression of liver fibrosis or HCC occurrence after SVR. This study revealed that neither the baseline LS value nor the degree of LS improvement was related to the poor outcomes. Similarly, previous prospective studies found no significant differences in HCC incidence among patients with a baseline LS ≥ 20 kPa and LS < 20 kPa, and that a 20% reduction in LS was not associated with a reduced risk of HCC and liver-related events [12,32]. By contrast, other studies reported a high LS value before DAA initiation was a risk factor for HCC [29,31]. A baseline LS of >21.3 kPa was found to be associated with a 4.2-fold increased risk of HCC [29]. Another study reported that the three-year estimated incidence of de novo HCC was 20% in patients with baseline LS ≥30 kPa compared to only 5% in patients with LS ≤ 30 kPa [31]. A retrospective study also reported that a 30% improvement in LS was inversely associated with the risk of HCC, but this association was not observed in our cohort [30]. The discrepancies in results could be explained by the differences in participant selection, with varying degrees of liver fibrosis and duration of the follow-up period.

Regarding liver steatosis, our study found no significant changes in CAP between baseline and any time points following SVR. This finding was in line with a previous study reporting that liver steatosis did not significantly improve after treatment and that the prevalence of hepatic steatosis in chronic HCV-infected patients after SVR with DAA was approximately 47% [33].

The current study illustrates the importance of monitoring LS after DAA therapy completion to predict the need for long-term follow-up for liver-related complications. However, the results of the study should be interpreted in light of its limitations. First, due to the retrospective design, each patient did not have LS measurement at every time point after SVR; therefore, comparing each pair of time points was not possible. Second, the data on alcohol consumption were incomplete or missing in the medical records of most patients; thus, the effect of alcohol use on the occurrence of liver-related events could not be evaluated. Third, the number of patients included in the study was relatively small, which potentially carries the risk of overestimating the magnitude of an association [34]. Nonetheless, the observed association was consistent with the majority of previous studies. Fourth, given the relatively short follow-up period, it was possible that some clinical events would have occurred later. Lastly, the number of patients with newly diagnosed HCC was insufficient to investigate the relationship between LS after SVR and HCC development. Further studies with more patients and a longer follow-up time are warranted to gain a better understanding of the liver-related outcomes caused by changes in LS after DAA therapy.

## 5. Conclusions

In summary, chronic hepatitis C patients who achieve SVR with DAA therapy have a significant improvement in LS. At 48 weeks after SVR, LS ≥ 8 kPa is significantly associated with the occurrence of liver-related events. Those HCV-infected patients who have LS ≥ 8 kPa at 48 weeks post-SVR should be closely monitored and given measures to prevent and detect liver-related complications.

## Figures and Tables

**Figure 1 medicina-59-00602-f001:**
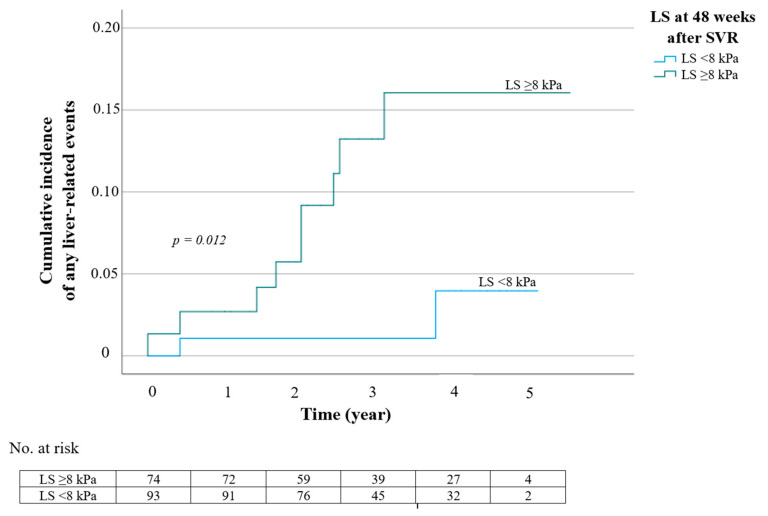
Cumulative incidence of any liver-related events according to liver stiffness (LS) at 48 weeks after sustained virological response (SVR).

**Table 1 medicina-59-00602-t001:** Baseline characteristics of study cohort.

Characteristics	Total n = 185	Characteristics	Total n = 185
Age (years) *	56.1 ± 10.9	HCV treatment status	
Male, n (%)	115 (62.2%)	Naïve HCV treatment, n (%)	130 (70.3%)
BMI (kg/m^2^) *	24.5 ± 3.9	Experienced treatment, n (%)	
Cirrhosis, n (%)	80 (43.2%)	PEG-IFN/RBV	52 (28.1%)
Comorbidity, n (%)		SOF-based regimen	3 (1.6%)
None	88 (47.6%)	HCV-related variables	
Hypertension	41 (22.2%)	HCV RNA (_log_IU/mL) *	5.9 ± 0.9
Dyslipidemia	13 (7.0%)	HCV genotype, n (%)	
Diabetes mellitus	35 (18.9%)	Genotype 1	100 (54.0%)
Chronic kidney disease	6 (3.2%)	Genotype 3	63 (34.0%)
Cardiovascular disease	7 (3.8%)	Genotype 6	22 (11.9%)
Cerebrovascular disease	3 (1.6%)	DAA regimen, n (%) ^†^	
Chronic respiratory disease	2 (1.1%)	SOF/DAC	35 (18.9%)
Others	44 (23.8%)	SOF/DAC/RBV	53 (28.7%)
Baseline laboratory results		SOF/LED	37 (20.0%)
Hemoglobin (g/dL) *	13.5 ± 1.9	SOF/LED/RBV	26 (14.1%)
White cell count (cell/mm^3^) *	6297 ± 2193	SOF/VEL	12 (6.5%)
Platelet count (×100/mm^3^) *	193 ± 87	SOF/VEL/RBV	6 (3.2%)
INR *	1.10 ± 0.15	SOF/IFN/RBV	11 (6.0%)
Alanine aminotransferase	70 (43–115)	Elbasvir/Grazoprevir	4 (2.2%)
(IU/L), median (IQR)		Duration of DAA regimen, n (%)	
Aspartate aminotransferase	56 (37–97)	12 weeks	175 (94.6%)
(IU/L), median (IQR) ^†^		16 weeks	1 (0.5%)
Serum albumin (g/dL) *	4.0 ± 0.4	24 weeks	9 (4.9%)
Total bilirubin (mg/dL) *	1.0 ± 0.7	Baseline LS (kPa), median (IQR)	11.8 (7.4–21.2)
Serum creatinine (mg/dL) *	0.8 ± 0.4	Baseline CAP (dB/m), median (IQR)	230 (204–259)
		Baseline FIB-4 index, median (IQR) ^†^	2.08 (1.21–4.0)

* Data are presented as mean ± standard deviation. ^†^ Two patients had missing data on baseline AST and FIB-4 index. One patient had missing data on DAA regimen. Abbreviations: BMI, body mass index; INR, international normalized ratio; LS, liver stiffness; CAP, controlled attenuation parameter; DAC, Daclatasvir; IFN, Interferon; LED, Ledipasvir; PEG-IFN, Pegylated interferon; RBV, Ribavirin; SOF, Sofosbuvir; TE, Transient elastography; VEL, Velpatasvir.

**Table 2 medicina-59-00602-t002:** Liver stiffness and liver steatosis values between baseline and 24, 48, 72, 96 weeks after SVR.

Time Point	Liver Stiffness Measurement			Liver Steatosis Measurement		
	n	Value (kPa), Median (IQR)	*p* *	n	Value (dB/m), Median (IQR)	*p* *
Baseline vs. 24 weeks after SVR	142	12 (7.8–21.5) vs. 8.5 (5.4–14.5)	<0.001	136	229 (200–259) vs. 239 (201–267)	0.410
Baseline vs. 48 weeks after SVR	34	8.5 (7.0–15.4) vs. 5.8 (4.8–10.5)	<0.001	32	222 (208–260) vs. 240 (202–266)	0.396
Baseline vs. 72 weeks after SVR	41	10.6 (7.3–21.3) vs. 6.1 (5.2–11.1)	<0.001	36	227 (215–267) vs. 244 (211–275)	0.818
Baseline vs. 96 weeks after SVR	48	10.8 (7.3–18.1) vs. 6.6 (4.8–12.0)	<0.001	36	224 (199–253) vs. 228 (206–254)	0.424

* Paired T-test. Abbreviation: SVR, sustained virological response.

**Table 3 medicina-59-00602-t003:** Rates of liver-related events occurring after successful treatment with direct anti-viral agents.

Outcome	Number of Events	Observational Period in Person-Years	Rate per 100 Person-Years
Any event *	12	508	2.36
Hepatocellular carcinoma	6	515	1.17
Portal hypertension-related liver decompensation (ascites, encephalopathy)	2	519	0.39
Listed for liver transplantation	1	528	0.19
Liver-related mortality	0	528	0.00
All-cause mortality	5	528	0.95

* Any event was a composite of all outcomes, in which the first event was counted if multiple events occurred in each patient.

**Table 4 medicina-59-00602-t004:** Demographics and liver stiffness values of six patients who developed HCC after SVR.

Number	Age	Sex	Presence of Cirrhosis before Treatment	Liver Stiffness at Baseline (kPa)	Liver Stiffness at 48 Weeks after SVR (kPa)	Time to Develop HCC after SVR
1	69	Female	Yes	21.8	19.4	10 months
2	66	Female	No	18.8	7.6	19 months
3	44	Male	Yes	17.5	15.4	29 months
4	57	Female	Yes	34.3	23.5	24 months
5	52	Male	Yes	35.3	21.8	20 months
6	83	Male	Yes	10.2	14.1	17 months

Abbreviations: HCC, hepatocellular carcinoma; SVR, sustained virological response.

**Table 5 medicina-59-00602-t005:** Univariate Cox proportional hazard analysis of factors associated with the incidence of any liver-related events.

Variables		HR (95%CI)	*p*
Age		1.08 (1.02–1.15)	0.008
Male sex		0.90 (0.28–2.83)	0.085
Body mass index		1.04 (0.90–1.21)	0.559
HCV Genotype	Genotype 3 Non-Genotype 3	0.96 (0.29–3.19) reference	0.944
History of HCV treatment	Experienced Naïve treatment	1.24 (0.39–3.91) reference	0.715
Cirrhosis status before treatment	Yes No	1.76 (0.56–5.56) reference	0.332
DAA treatment	RBV-containing regimen RBV-free regimen	0.45 (0.48–5.29) reference	0.452
Liver stiffness at baseline	≥16 kPa <16 kPa	1.71 (0.55–5.30) reference	0.353
Liver stiffness at 48 weeks after SVR *	≥8 kPa <8 kPa	5.68 (1.23–26.31) reference	0.026
Decreased liver stiffness value from baseline to 48 weeks after SVR	≤3 kPa >3 kPa	2.98 (0.79–11.24) reference	0.107
Baseline FIB-4 index at baseline	≥3.50 <3.50	2.46 (0.76–7.99) reference	0.134
FIB-4 index at 48 weeks after SVR	≥1.35 <1.35	5.18 (1.10–24.32) reference	0.037
Decreased FIB-4 index from baseline to 48 weeks after SVR	≤1.30 >1.30	1.88 (0.58–6.09) reference	0.291
Baseline CAP	≥230 dB/m <230 dB/m	0.33 (0.09–1.21) reference	0.095

Abbreviations: DAA, direct antiviral antigen; RBV, Ribavirin; LS, liver stiffness; CAP, controlled attenuation parameter; SVR, sustained virological response HR, hazard ratio. * LS at 48 weeks after SVR in this part included LS at 24 weeks before and after this time point, i.e., LS at 24 and 72 weeks after SVR were used in patients who had not conducted TE at 48 weeks after SVR. A total of 169 patients were included.

**Table 6 medicina-59-00602-t006:** Multivariate Cox proportional hazard analysis of factors associated with the incidence of any liver-related events.

Variables	Model 1		Model 2		Model 3	
	Adjusted HR (95%CI)	*p*	Adjusted HR (95%CI)	*p*	Adjusted HR (95%CI)	*p*
Age	1.07 (1.00–1.14)	0.025	1.06 (0.99–1.13)	0.110	1.06 (0.99–1.13)	0.120
Male sex	1.32 (0.39–4.53)	0.655	1.23 (0.36–4.22)	0.74	1.47 (0.39–5.60)	0.569
Liver stiffness at 48 weeks after SVR ≥8 kPa <8 kPa	5.06 (1.09–23.42) reference	0.038			5.04 (1.01–25.26) reference	0.049
FIB-4 index at 48 weeks after SVR≥1.35 <1.35			3.08 (0.56–16.97) reference	0.198	1.91 (0.33–11.10) reference	0.468

Abbreviations: SVR, sustained virological response; HR, hazard ratio.

**Table 7 medicina-59-00602-t007:** Observational studies assessing changes in liver stiffness post SVR and its association with HCC incidence in chronic HCV patients.

Study, Year	Enrollment Year (N)	Treatment and Duration	Baseline TE, Median (IQR), kPa	TE post-SVR, Median (IQR), kPa	Follow-up Duration	Incidence of HCC Post-SVR
Conti, 2016 [29]	2015 (344)	SOF/SMV, SOF/RBV SOF/DAC, SOF/LED DAC/SIM, 3D Duration—NA	23.6 ± 10.4 (mean ± SEM)	mean TE ± SEM was 23.1 ± 0.8 vs. 28.1 ± 2.5 in patients without vs. with HCC	24 weeks	17/344 (4.94%) Follow-up LS > 21.3 kPa was associated with 4.2-fold increased HCC risk
Kanwal, 2017 [25]	2015 (22, 500)	SOF/LED, SIM PAR/Ritonavir, DAC Duration—NA	NA	NA	NA	0.90/100 patient-years
Ravaioli, 2018 [30]	2015–2016 (139)	SOF, SOF/DAC SOF/SMV, SOF/LDV PAR/r/OBV+DAS PAR/r/OBV 12 or 24 weeks	18.6 (15–26.3)	EOT: 13.8 (10.4–20.4)	15 months	20/139 (14.4%)
Degasperi, 2019 [31]	2014–2016 (565)	SOF/LED, SOF/RBV SOF/DAC, SOF/SIM O/P/R/DAS, O/P/R 12 weeks (56%), 24 weeks (36%), 16–20 weeks (8%)	19.1 (12.0–75.0)	NA	25 months (range 3–39)	The three-year cumulative incidence of HCC was 20% vs. 5% in patients with baseline LS > 30 kPa vs. ≤ 30 kPa
Pons, 2019 [12]	2015–2016 (572)	SOF/SIM, SOF/LED SOF/DAC, PTV/r/O/D Others 12 weeks	20.2 ± 10.4 (mean ± SD)	SVR48: 13.9 ± 9.2 (mean ± SD)	2.8 years	1.5/100 patient-years
Morisco, 2021 [32]	2015–2017 (706)	SOF, SOF/other, 3D, 2D Duration—NA	NA	NA	28 months ± 5 (mean ± SD)	1.6/100 patient-years
Shiha, 2022 [10]	2015–2019 (1517)	Types of DAA - NA 12 or 24 weeks	12.4 (11.1–14.2)	EOT: 8.8 (6.6–12.3)	28 months (range 24–32)	0.91/100 patient-years
The present study	2016–2020 (185)	SOF/DAC, SOF/DAC/RBV, SOF/LED SOF/LED/RBV, SOF/VEL SOF/VEL/RBV, SOF/IFN/RBV; 12, 16, or 24 weeks	11.8 (7.4–21.2)	SVR24: 8.5 (5.4–14.) SVR48: 5.8 (4.8–10.5) SVR72: 6.1 (5.2–11.1) SVR96: 6.6 (4.8–12.0)	41.6 months	6/185 (3.24%) Follow-up at 48 weeks after SVR LS ≥ 8kPa was associated with liver-related events (HR = 5.68)

Abbreviations: 3D, Ombitasvir, Paritaprevir with Ritonavir, and Dasabuvir; DAC, Daclatasvir; EOT, end of treatment; LED, Ledipasvir; NA, not available; PAR/r/OBV+DAS, Paritaprevir/Ritonavir/Ombitasvir/Dasabuvir; PTV/r/O/D, Paritaprevir/Ritonavir/Ombitasvir/Dasabuvir; RBV, Ribavirin; SIM, Simeprevir; SOF, Sofosbuvir; SVR (number), weeks after SVR; VEL, Velpatasvir.

## Data Availability

The data of this study are available upon request.

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
