# Peer review of "Association between Liver Stiffness and Liver-Related Events in HCV-Infected Patients after Successful Treatment with Direct-Acting Antivirals"

_medicina, 2023, doi:10.3390/medicina59030602_

Round 1

Reviewer 1 Report

This manuscript is an interesting study on the follow-up of patients with hepatitis C who have been treated with direct-acting antivirals. The authors look for relationships between hepatic events in a cohort that might be related to changes in liver stiffness. Presented data is interesting and there are some correlations found, most relevant being the cut-off value of LS ≥8 kPa at 40 weeks post-SVR associated with any liver-related event. Of course, the study has limitations, which have been well reported by the authors in the discussion. One of them is especially important, the limited number of patients analyzed.

The most prominent issue that should be corrected is the lack of information about ethical aspects, such as the registration of the protocol in an ethics committee, as well as establishing whether informed consent was obtained from the patients.

Other questions:

What do you mean "5 patients died from non-liver-related causes"?

Reviewer 2 Report

This study showed: patients with LS ≥8 kPa at 48 weeks after SVR should be regularly monitored for liver-related 30 complications, particularly HCC development.

However, I had some questions:

1.How was the ratio of alcohol user? And the amount of alcohol use?

2.How was the FIB4 change among this patients? Is the LS post-SVR status had better prediction than FIB4? or GGT?

3.Can you draw KM curve for the events?

4. Is there any HBV co-infection?

5.The cirrhosis patients acounted for 40%. How about the  LS ≥8 kPa at 48 weeks after SVR affecting the result in cirrhotic and non-cirrhotic patients, respectively?

Round 2

Reviewer 2 Report

.